# Development of Indirect Competitive ELISA and Colloidal Gold Immunochromatographic Strip for Endosulfan Detection Based on a Monoclonal Antibody

**DOI:** 10.3390/foods12040736

**Published:** 2023-02-08

**Authors:** Xinghua Zhou, Shuoning Guan, Na Li, Jiayu Zhou, Wenwen Pan, Yun Wang

**Affiliations:** School of Food and Biological Engineering, Jiangsu University, Zhenjiang 212013, China

**Keywords:** endosulfan, monoclonal antibody, ic-ELISA, colloidal gold, immunochromatographic strip

## Abstract

Endosulfan, as an effective broad-spectrum insecticide, has been banned in agricultural areas because of the potential harmful effects on human health. This study aimed to develop an indirect competitive enzyme-linked immunosorbent assay (ic-ELISA) and colloidal gold immunochromatographic (ICA) strip based on a prepared monoclonal antibody (mAb) for quantitative and qualitative detection of endosulfan. A new mAb with high sensitivity and affinity was designed and screened. The ic-ELISA showed a 50% inhibition concentration (IC_50_) value of 5.16 ng/mL for endosulfan. Under optimum conditions, the limit of detection (LOD) was determined to be 1.14 ng/mL. The average recoveries of endosulfan in spiked pear and apple samples ranged from 91.48–113.45% and 92.39–106.12% with an average coefficient of variation (CV) of less than 7%, respectively. The analysis of colloidal gold ICA strip could be completed within 15 min by naked eye and the visual limit of detection (vLOD) was both 40 ng/mL in pear and apple samples. In conclusion, both developed immunological methods were suitable and reliable for the on-site detection of endosulfan in real samples at trace levels.

## 1. Introduction

Endosulfan is a member of the organochlorine pesticide that is used to control insects and pests on crops, including fruits, vegetables and cereal [1]. It is a mixture of α-endosulfan and β-endosulfan with a ratio of 7:3 [2]. Endosulfan can obstruct the γ amino-butyric acid (GABA)-gated chloride channel, thereby disrupting the normal nerve function of insects and pests [3]. Some studies indicated that endosulfan is an endocrine disruptor, causing reproductive and heredity toxicity in human and animals [4]. Moreover, endosulfan was classified as a persistent organic pollutant (POP) with bioaccumulation, a high toxicity and potential carcinogenicity. It will have a long-term harmful impact on the environment and on humans [5]. In addition, endosulfan was added to the Stockholm Convention’s list of prohibited substances in 2011 and has been roundly forbidden for use in agricultural areas in China since 2019. Therefore, developing a reliable and rapid method to monitor endosulfan residue in food is of great importance. 

Currently, traditional detection methods for endosulfan residue include gas chromatography-mass spectrometry (GC-MS) [6,7], liquid chromatography–tandem mass spectrometry (LC-MS) and high performance liquid chromatography (HPLC) [8,9,10]. The above methods have a high sensitivity and accuracy, but they are generally time-consuming and require expensive equipment, a professional technician and tedious extraction. Therefore, novel, rapid and accurate on-site analytical methods are required for endosulfan detection in food. 

An immunoassay is based on the specific reaction between the antibody and antigen, and is widely used to detect pesticide, veterinary drugs [11,12], fungal toxin and food-borne pathogens [13,14]. Compared with instrumental analytical methods, the immunoassay is rapid, has a low cost, simple extraction and operation without professional operators. However, there are few reports about the immunoassay on the detection of endosulfan residue. Wang et.al developed an ic-ELISA for endosulfan based on the polyclonal antibody (pAb) [15], which was very different in batches and had a poor stability. Deepak et.al developed an ic-ELISA for α-endosulfan and β-endosulfan detection [16], but the sensitivity and selectivity of the method were poor due to high IC_50_ for α-endosulfan (79.4 ng/mL) and β-endosulfan (86.21 ng/mL). A colloidal gold immunochromatographic (ICA) strip, an instrument-free method, is simpler and faster, compared with ELISA. In addition, it is used for the on-site screening of large numbers of samples rapidly. At present, there is only one report on colloidal gold ICG strips for the detection of endosulfan residue. Karanthi et.al reported that a colloidal gold ICA strip was developed for endosulfan, and the limit of detection (LOD) was 1800 μg/L [17], which cannot satisfy the standards for residue limits. Therefore, it is significant to develop a new ic-ELISA and ICA strip for the quantitative and qualitative detection of endosulfan.

In this study, the monoclonal antibody (mAb) with a high sensitivity and affinity against endosulfan was prepared, and then an ic-ELISA and colloidal gold ICA strip were established based on mAb for the rapid determination of endosulfan.

## 2. Materials and Methods

### 2.1. Materials and Instruments

Endosulfan, toxaphene, chlordane, heptachlor, hypoxanthine aminopterin, bovine serum albumin (BSA), ovalbumin (OVA), carbodiimide (EDC), Freund’s complete adjuvant, Freund’s incomplete adjuvant, chloroauric acid (HAuCl3·4H_2_O), trisodium citrate, thymidine medium (HAT) and hypoxanthine-thymidine medium (HT) were purchased from Sigma-Aldrich (St. Louis, MO, USA). Horseradish peroxidase (HRP)-conjugated goat anti-mouse IgG, 3,3′,5,5′-tetramethylbenzidine (TMB) and Freund’s complete and incomplete adjuvant were acquired from Sangon Biotech Co., Ltd. (Shanghai, China). Immunogen and coating antigen were prepared in our laboratory. All other reagents were acquired from the National Pharmaceutical Group Chemical Reagent Co., Ltd. (Shanghai, China). 

The microplate reader (1201–7044) was obtained from Thermo Fisher Scientific (Shanghai, China) Co., Ltd. (USA). The CT300 CNC strip cutting machine, HM3030 XYZ three-dimensional dispensing platform, ZQ3500 CNC fast chopping machine were obtained from shanghai Kinbio Tech Co., Ltd. (Shanghai, China). The gas chromatography-mass spectrometer (GC-MS) was obtained from Thermo Fischer Scientific (USA).

### 2.2. Endosulfan Hapten Synthesis and Identification

We Weighed 35.7 mg of mercaptan into a 5 mL single port bottle and added 1 mL of pyridine. into a dissolution well. Twelve mg of succinic anhydride and 2 mg of DMAP were sequentially added and stirred at 50 °C. Following the 6 h reaction, the solvent was rotary evaporated, 0.5 mL pure water was added to dissolve the precipitate, the above solution pH was adjusted to 10 with an NaOH solution (1 M), extracted with 5 mL n-hexane. All of the aqueous phases were collected and the aqueous solution pH was adjusted to 4 with an HCl solution (1 M), there was the solid evolution and the target was obtained after filtration and drying (the synthesis process is shown in Figure 1). Finally, we identified with MS whether the derivatization was successful.

### 2.3. Synthesis of Endosulfan Complete Antigen

Endosulfan complete antigen was synthesized by the activated ester method, and the reaction is shown in the Figure 2. Three mg hapten was dissolved in 0.5 mL DMF. Next, 2.2 mg NHS and 4 mg EDC were added to the hapten solution, and the mixture was produced by a chemical reaction at room temperature following 6 h under dark conditions. Fifteen mg BSA and 10 mg OVA were weighed and dissolved in 6 mL CBS buffer, the activated endosulfan hapten solution was added dropwise in the protein solution under magnetic stirring, and left to react overnight at room temperature. Dialysis with a phosphate (PBS) buffer at 4 °C was conducted for 72 h. The dialysate was changed regularly during this period; it was aliquoted after dialysis, then stored at −20 °C for later use.

### 2.4. Preparation of the mAb against Endosulfan

The mAb was prepared using the hybridoma technique [18]. Five female BALB/c mice (6–8 weeks of age) were injected subcutaneously with immunogen at multiple sites on the back. The dose of the initial immunization was 100 µg of immunogen, which was diluted in physiological saline and emulsified with equal volume of Freund’s complete adjuvant per mouse. Then four booster immunizations were given triweekly intervals, using 50 µg of immunogen diluted in physiological saline and emulsified with equal volumes of Freund’s incomplete adjuvant per mouse. The sera of each mouse were analyzed by ic-ELISA, and the mouse with the highest affinity and inhibition was selected for cell fusion. The SP2/0 myeloma cells were mixed with spleen cells at ratio of 10:1 and fused with 1 mL of PEG1500. Then, the fused cells were cultured in HAT medium, and eight days after cell fusion, the HAT medium was replaced by HT medium. The positive wells were screened by ic-ELISA, and sub-cloning was conducted three times by limiting the dilution method. The hybridoma cells that showed the best affinity and inhibition were intraperitoneally injected into the mice for mAb production, then the ascites were collected and purified by ammonium sulfate precipitation [19].

The isotype of mAb was determined by the isotype kit, the affinity constant (Ka) was determined with coated antigen and mAb dilution method by ELISA [20].

### 2.5. Development of ic-ELISA

The preparation of the ic-ELISA kit has been described elsewhere [21]. The 96-well plate was coated with antigen (100 μL per well) and incubated at 37 °C for 2 h. The plate was then washed 3 times with PBS (250 μL/well) containing 0.05% Tween-20 and closed with CBS (200 μL/well) containing 0.2% gelatin and incubated at 37 °C for 2 h. Then, 50 μL of standard solution and 50 μL of a monoclonal antibody were added sequentially to each well and incubated at 37 °C for 30 min. Following three washes, HRP-labelled goat anti-mouse IgG was added to each well (100 µL per well) and incubated at 37 °C for 30 min. Plates were washed once more (three times) and the substrate (TMB) solution (100 µL per well) was added and incubated at 37 °C for 15 min. Then, the reaction was terminated by 2 M H2SO4 (50 µL per well). Finally, the optical density (OD) was measured at 450 nm with a microplate reader.

### 2.6. Optimization of ic-ELISA

The ic-ELISA performance was mainly affected by the standard dilution buffer [22], including pH values (6.0, 7.2, 8.5, 9.6), organic solvents (0%, 5%,10%, 15%, 20%) and ionic strength (0%, 0.4%, 0.8%, 1.6%, 3.2%). To improve the sensitivity of ic-ELISA, the above parameters were optimized, and the standard curves were established under various conditions. IC_50_ and A_max_ (absorbance of zero concentration at 450 nm) were used to assess the performance of mAb in ic-ELISA [23].

### 2.7. Specificity Determination

The specificity of mAb was assessed by a cross-reactivity experiment. Structural analogs were studied to determine the IC_50_ value using the optimized ic-ELISA procedure described above. The cross-reactivity (CR%) of mAb was calculated using the following formula:$$\mathrm{CR}\;(\%)=(\frac{{\mathrm{IC}}_{50}\;\mathrm{of}\;\mathrm{endosulfan}}{{\mathrm{IC}}_{50}\;\mathrm{of}\;\mathrm{analogues}})\;\times \;100\%$$

### 2.8. Preparation of Colloidal Gold Labelled mAb

The colloidal gold was synthesized, as previously described [24]. In brief, 2.5 mL of 1% freshly prepared trisodium citrate (*m*/*v*) was added to 50 mL boiling 0.01% HAuCl_4_ (*m*/*v*) solution with gentle stirring, and refluxed for 30 min. Once cooled down, the colloidal gold solution was stored at 4 °C for future use. The resulting colloidal gold was characterized by UV–vis spectrum and transmission electron microscopy (TEM).

The colloidal gold was labelled with mAb based on the reported literature [25]. Firstly, the pH of the colloidal gold solution (3 mL) was adjusted to 8.5 with 0.1 M K_2_CO_3_, and then the prepared mAb (0.15 mg) was added dropwise to the above solution under gentle stirring for 40 min at room temperature. The 600 µL of 10% BSA (*w*/*v*) was slowly added to block any unreacted sites for 30 min, then the mixture was centrifuged at 7000 rpm for 15 min. Finally, the precipitate of the colloidal gold-labelled mAb was resuspended in PBS (0.01 M pH 7.2) containing 2% (*w*/*v*) BSA, 1% (*w*/*v*) sucrose, 0.5% Tween-20 and 0.02% (*w*/*v*) sodium azide and stored at 4 °C in the dark for future use.

### 2.9. Preparation and Principle of the Colloidal Gold ICA Strip

The colloidal gold ICA strip was developed, as previously reported [24]. The assembly and structure of the colloidal gold ICA strip are shown in Figure 3A, the nitrocellulose membrane (NC membrane) was pasted onto the middle of the PVC backing card, the absorbent pad and sample pad were adhered to the upper and lower sides of the PVC backing card, respectively, with an overlap on the NC membrane by 2 mm approximately. Prior to assembling the latera-flow ICA strip, the sample pad (glass fiber membrane) was saturated with 0.01 M PBS (pH 7.4) containing 0.2% (*v*/*v*) Tween-20, 1.0% (*w*/*v*) sucrose and 1.0% (*w*/*v*) BSA to reduce the matrix interference and was then dried at 37 °C. The coating antigen (0.8 mg/mL) and goat anti-mouse IgG (0.4 mg/mL) were sprayed on the NC membrane as the test line (T line) and control line (C line), respectively, and the distance between the T line and C line was 8 mm. Finally, the card was cut into 2 mm wide individual strips after drying at 37 °C and then stored in a desiccator for future use.

As shown in Figure 3B, 150 µL of the sample extraction solution was added to the well. The solution was mixed evenly with pipettor and incubated for 5 min at room temperature, and then the strip was inserted into the well. Due to the capillary action, the mixture was migrated from the sample pad to the absorbent pad, captured by the coating antigen on the T line and the goat anti-mouse IgG on the C line was based on the antibody-antigen reaction. Then, after 3 min, the detection results could be visualized by the naked eye.

### 2.10. Parameter Optimization of the Dipstick

#### 2.10.1. Selection of the Labeled Antibody pH

The pH of the best labeled antibody in the assay was adjusted with 0.1 M K_2_CO_3_. Seven centrifuge tubes were taken and 1 mL of the colloidal gold solution was added, followed by 0, 2, 4, 6, 8, 10 and 12 μL of the 0.1 M K_2_CO_3_ solution. The solution was mixed well and 10 μL of monoclonal antibody prepared at 0.5 mg/mL was added. Again, this was mixed well and then allowed to stand for 30 min to observe the color change of the solution. The colloidal gold solution with no change in color was selected, and the one with the least amount of K_2_CO_3_ solution was added.

#### 2.10.2. Selection of the Amount of the Labeled Antibody

To select the amount of the optimized K_2_CO_3_ solution for the above tests, 1 mL of the colloidal gold solution was added to 8 centrifuge tubes, followed by 2, 3, 4, 5, 6, 7, 8 and 9 µg of antibodies, respectively, sequentially, and the color change was observed after standing for 5 min at room temperature. The set with the least amount of antibody added was selected with the premise that color remained constant.

#### 2.10.3. Selection of the T-Line Coating Concentration

The coatings of 0.2, 0.4, 0.8, 1.2, 1.6 and 2.0 mg/mL were sprayed on the T-line, respectively, and the color development reactions were performed after drying, and the best color development group was selected as the optimal concentration of the coating original.

### 2.11. Sample Analysis

The homogeneous pear and apple samples known to be endosulfan free (5 g) were added to 50 mL centrifugal tubes with various concentrations of endosulfan, and 5 mL of methyl alcohol were added to the tube. The mixture was vibrated for 5 min vigorously and then filtrated. The filter liquor was collected and transferred into tubes which were analyzed by the ic-ELISA and the colloidal gold ICA strip.

In addition, samples were simultaneously analyzed by GC-MS to confirm the validity of the bands. GC-MS working conditions refer to SN/T 1873-2019 (method for detection of endosulfan residues in exported foods).

## 3. Results and Discussion

### 3.1. Identification of the Endosulfan Haptens

Figure 4 is the MS identification result of the endosulfan haptens, in negative ion mode, the molecular mass of the target is 459, in keeping with the molecular mass of the endosulfan haptens (458), indicating a successful endosulfan hapten derivation.

### 3.2. Identification of the Endosulfan Complete Antigen

In this test, endosulfan immunogen and coating Gen were prepared by the derivatization of endosulfanol with a similar structure to endosulfan using the succinic anhydride method to obtain carboxyl groups that can directly react with proteins, and then these were coupled with BSA and OVA, respectively. Following derivatization, there was no UV absorption peak on the structure of the endosulfan haptens, so a native PAGE was used to characterize it, and the results are shown in Figure 5. The separation of proteins is not only related to the molecular weight, but also to the charge carried by the protein. The net charge distribution of the BSA is relatively uniform with clear bands. When and haptens are coupled, the net charge distribution is not uniform, a backward extension appears, and the band broadens. Indicating the successful conjugation of the BSA by the endosulfan haptens, i.e., the successful endosulfan immunogen preparation. The appearance of an anterior posterior extension with broadening of the bands compared to the coater’s strips and OVA may also indicate a successful coater preparation. Finally, the production of endosulfan antibodies could also prove to be a successful endosulfan immunogen and coating antigen coupling.

### 3.3. Screening of the Hybridoma Cell

The five BALB/c mice were injected by the prepared immunogen 5 times, while the antisera were determined by ic-ELISA after the third immunization. The results are shown in Table 1. The M-1 with the highest titer and inhibition was selected for cell fusion. Spleen cells and SP2/0 were mixed at ratio of 10:1 and fused by PEG, among which the positive hybridomas named 1D6, 2C8, 3B4, 6C5 and 7H2 were obtained. Next, the ic-ELISA was applied to determine the inhibition and titer, while the optimal hybridoma was selected to prepare the ascites and mAb (Figure 6). The A_max_/IC_50_ of 1D6, 3B4 and 6C5 hybridoma were better than 2C8 and 7H2 hybridoma The A_max_ of 1D6 hybridoma was the highest, while the A_max_/IC_50_ and IC_50_ of 3B4 hybridoma was the lowest among the five hybridomas. In short, the 3B4 hybridoma was selected to prepare the ascites and to obtain mAb against endosulfan.

### 3.4. Characterization of mAb against Endosulfan

According to the instructions of the isotype kit, the HRP-IgG was replaced with the different antibody isotype reagents, and the following operation was the same as the ELISA. The results are showed in Figure 7A, the heavy and light chain were IgG1 and Lambda, respectively. Therefore, the ascites were purified by the ammonium sulfate precipitation. Three coating antigen concentrations of 0.3, 0.1 and 0.003 µg/mL were selected to determine the affinity constant of mAb, according to the following formula: Ka = (n − 1)/2(n [Ab] t − [Ab] t), and the result in Figure 7B showed that the affinity constant of mAb was 3.80 × 10^9^ L/mol, which indicated that the antigen could bind tightly to the antibody.

### 3.5. Development and Optimization of the ic-ELISA

The ic-ELISA was optimized under the optimal analysis condition, with the coating antigen and mAb concentrations for 0.1 μg/mL and 0.31 μg/mL, respectively. The pH values were known to affect the available number of sites of the antigen-antibody reaction, while the NaCl content could affect the charge of the epitope and paratope groups and then limit the antigen-antibody reaction. As endosulfan was extracted by methyl alcohol, the sensitivity of the ic-ELISA would be affected by the solvent. As shown in Figure 8 and Table 2, this optimum condition was reached at pH 7.2, which gave the lowest IC_50_ value of 5.55 ng/mL, and the highest A_max_/IC_50_ ratio of 0.27. 5% methyl alcohol was selected as the ideal condition with the lowest IC_50_ value of 5.67 ng/mL and with the highest A_max_/IC_50_ ratio of 0.27. With regard to the NaCl content, the IC_50_ was not the lowest value at 0% for the NaCl content, but it gave the highest A_max_ value and the highest A_max_/IC_50_ ratio. In short, the optimum conditions for endosulfan through the ic-ELISA were a pH of 7.2, 5% methyl alcohol content and 0% NaCl content.

The standard curve of endosulfan in the ic-ELISA was established under the optimal condition. As shown in Figure 8D, the equation was y = 0.135 + 1.328/[1 + x/5.160^0.918^] and the linear regression correlation coefficient (R^2^), the IC_50_, the linear range (IC_20_–IC_80_), the limit of detection (LOD, IC_20_) were 0.99, 5.16 ng/mL, 1.14–23.38 ng/mL, 1.14 ng/mL, respectively [26]. In all, the sensitivity of antibody prepared in the paper was better than in the reported paper [15].

### 3.6. Specificity of mAb

As shown in Table 3, the crossover rates of the antibodies and other classes were low. It is generally accepted that a crossover rate of less than 5% can be considered as no crossover, and the crossover rate with heptachlor was 3.56% and less than 0.1% for toxaphene and chlordane, so the endosulfan mAbs produced in this assay have a high specificity.

### 3.7. Preparation of the Colloidal Gold Labelled mAb

The quality of the colloidal gold plays an important role in the sensitivity and stability of the colloidal gold ICA strip. The prepared colloidal gold was assessed by UV–vis spectrum and transmission electron microscopy (TEM) images. In the UV–vis spectrum of colloidal gold (Figure 9A), the maximum absorption wavelength for colloidal gold was 528 nm, which indicated that colloidal gold was successfully prepared. The TEM of colloidal gold is shown in Figure 9B, the shape and size of the colloidal gold was homogeneous, with a good dispersity and no aggregation, and the average diameter of the colloidal gold was 13.14 ± 4.75 nm. Therefore, the mAb with a positive charge must be conjugated to GNPs with negative charge firmly via electrostatic attraction. The prepared colloidal gold was consistent with the reported paper [27], which indicated that the colloidal gold could be applied in colloidal gold ICA strip.

### 3.8. Determination of the Parameters for the Dipstick Optimization

Under weak base conditions, colloidal gold is more favorable and antibody binding. The K_2_CO_3_ solution was chosen in this test to adjust the pH in the environment to see if the color of the solution changes and if the color of the solution starts to change, it indicates that the colloidal gold is unstable and aggregation occurs under this condition. At an optimal pH, we determined the optimal labeling amount of the antibody by observing the solution color change. In this test, by naked eye, the optimal conditions of the endosulfan colloidal gold test paper strips were as follows: 1 mL colloidal gold, 8 μL K_2_CO_3_ solution and 8 µg of the antibody.

Under optimal conditions of the pH and the amount of labeled antibody, the coating stock was diluted into different concentrations, sprayed on the NC membrane, and the color reaction was performed. The optimal coating stock concentration for the endosulfan colloidal gold test paper strips by color reaction was 0.8 mg/mL, respectively.

### 3.9. Ic-ELISA and the Colloidal Gold ICA Strip for Endosulfan Detection

To test the feasibility of our developed method, the samples were fortified with different concentrations of endosulfan and analyzed by the developed ic-ELISA and the colloidal gold ICA strip.

In the ic-ELISA method, the recovery and the coefficient of variation (CV) were used to determine the accuracy and precision of the assay (Table 4). The recoveries of the pear samples were 91.48–113.45%, while the recoveries of the apple samples were 92.39–106.12%. The average CV of the pear and apple samples was less than 7%. The above results demonstrated that the ic-ELISA was accurate and repeatable, which could be applied for the quantitative detection of endosulfan in real samples.

In the colloidal gold ICA strip, the visual limit of detection (vLOD) was defined as the minimum concentration showing a weaker T line compared with the C line color. As shown in Figure 10, the color of the T line was much more shallow when the concentration of endosulfan increased. The vLOD of the colloidal gold ICA strip was 40 ng/mL in pear and apple samples. Kranthi has developed the lateral-flow immunoassay based on colloidal gold for endosulfan detection with LOD 1800 ng/mL [17], which was more sensitive than the colloidal gold ICA strip in this paper and could not satisfy the national maximum residue limits (MRL). The above results indicated that the developed colloidal gold ICA strip was suitable and accurate for the on-site detection of real samples within 15 min in an equipment-free analysis.

Figure 10 shows the test results of the dipstick, and by comparison with Table 5 (GC-MS analysis of pear and apple spiked samples), it shows consistency between the test and the instrumental analysis of the dipstick in this concentration range. It illustrates the accuracy of this paper’s test strips. Comparing this method with other methods in terms of detection limits and time, Table 6 (Comparison of this method with other measurement methods) shows that our established method is sensitive, fast and simple.

## 4. Conclusions

The 3B4 hybridoma was screened to prepare ascites, and the mAb against endosulfan was obtained with 3.80 × 10^9^ L/mol constant affinity. The highly sensitive ic-ELISA and colloidal gold ICA strip based on the mAb were developed for the detection of endosulfan in pear and apple samples. The recovery rates for the ic-ELISA of the pear and apple were 91.48–113.45% and 92.39–106.12%, respectively, and the average CV for both was less than 7%. In the colloidal gold ICA strip, the vLOD was 40 ng/mL for the pear and apple samples. In conclusion, the results indicated that the developed method was sensitive, rapid and applicable for the on-site detection of endosulfan.

## Figures and Tables

**Figure 1 foods-12-00736-f001:**
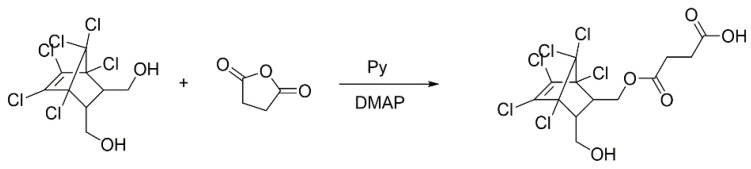
Synthesis of the endosulfan haptens.

**Figure 2 foods-12-00736-f002:**
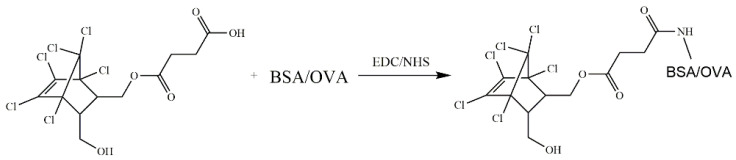
Synthesis of the immunogen antigen and coating antigen of endosulfan.

**Figure 3 foods-12-00736-f003:**
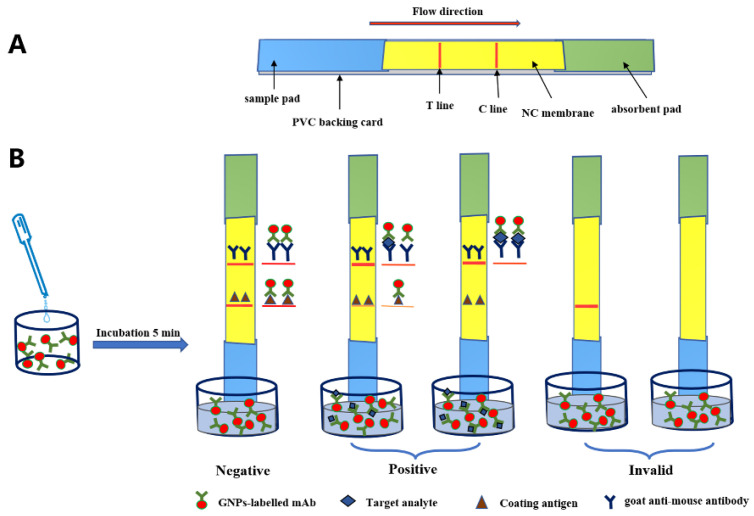
Illustration of the colloidal gold ICA strip: (**A**) structure and (**B**) principle.

**Figure 4 foods-12-00736-f004:**
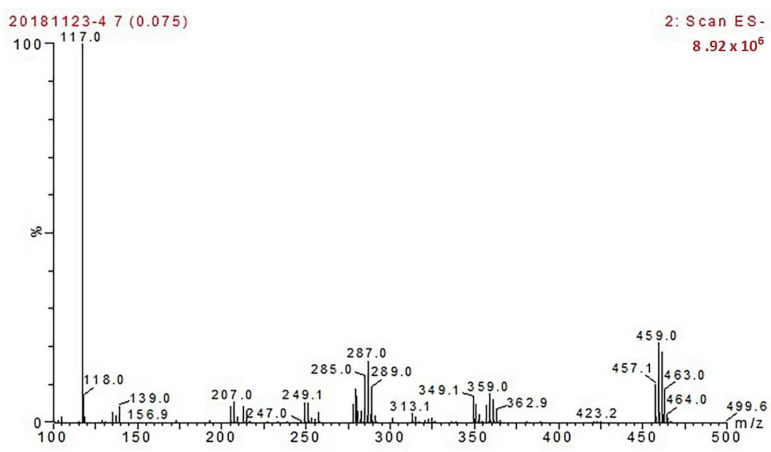
Identification of the endosulfan haptens by MS.

**Figure 5 foods-12-00736-f005:**
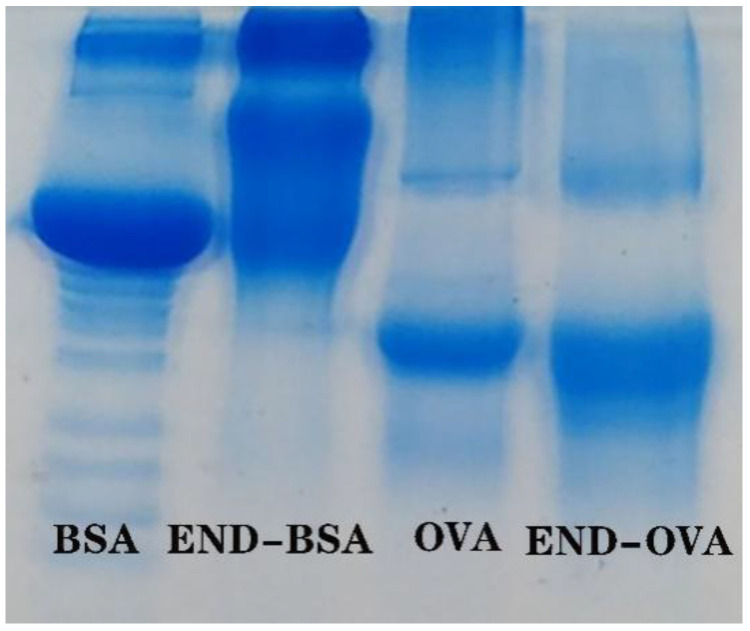
Native PAGE of immunogen and coating antigen of endosulfan.

**Figure 6 foods-12-00736-f006:**
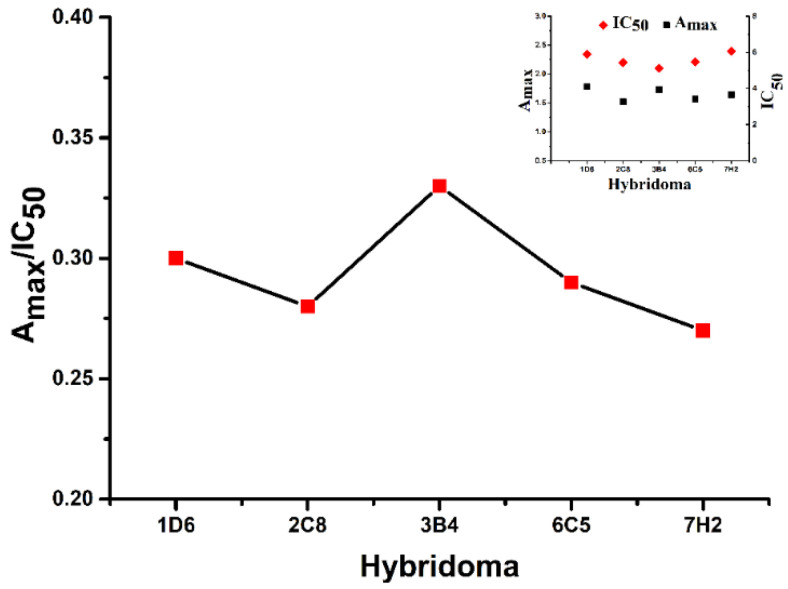
The screening results of hybridoma against endosulfan.

**Figure 7 foods-12-00736-f007:**
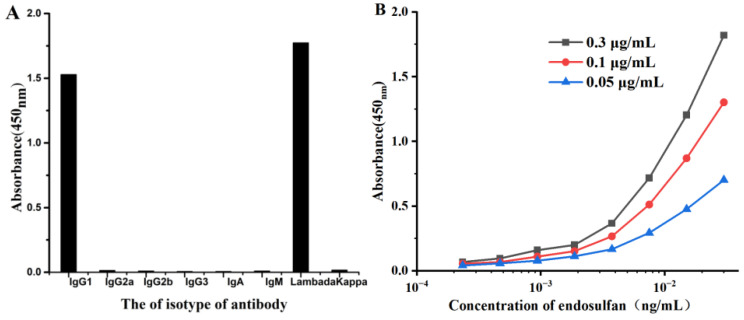
Characterization of the hybridoma antibody. (**A**) Isotype determination of mAb. (**B**) Affinity constant result of mAb.

**Figure 8 foods-12-00736-f008:**
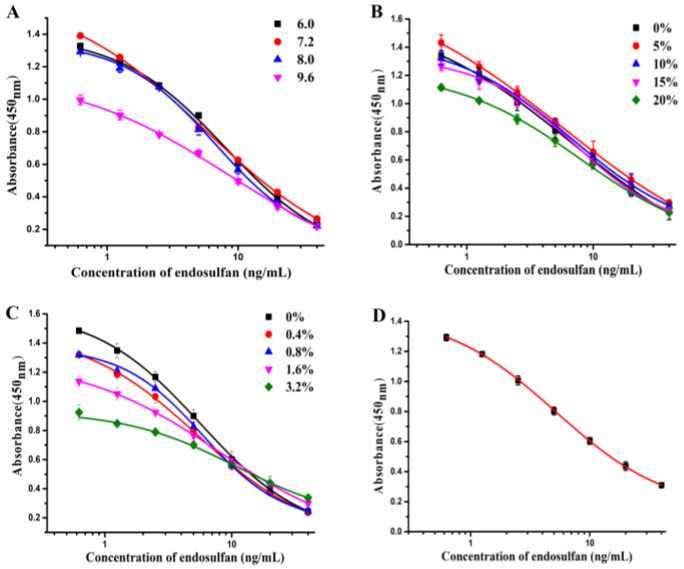
The optimization of the ic-ELISA. (**A**) pH values; (**B**) methyl alcohol content; (**C**) NaCl content; (**D**) The standard curves of endosulfan using the ic-ELISA. (*n* = 3).

**Figure 9 foods-12-00736-f009:**
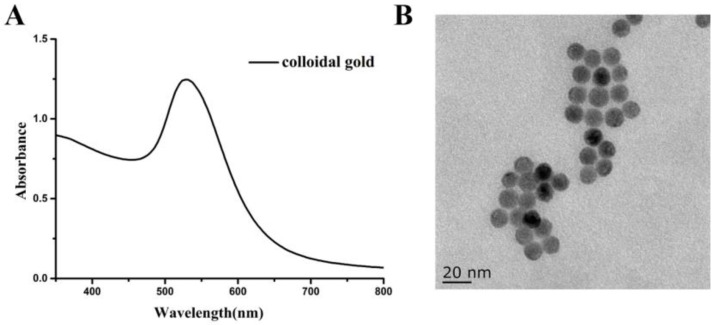
The characterization of colloidal gold: (**A**) UV-vis spectrum and (**B**) TEM.

**Figure 10 foods-12-00736-f010:**
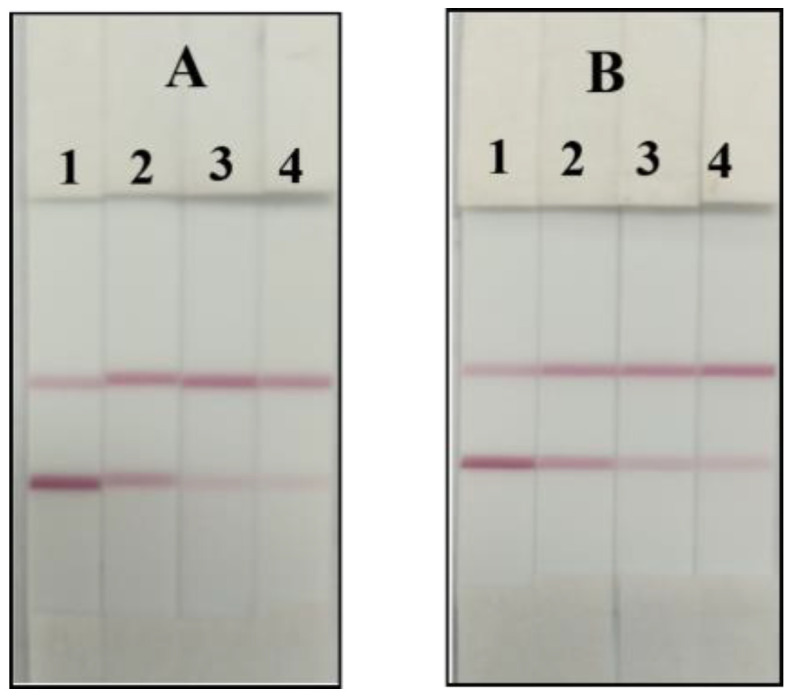
The sensitive analysis of the endosulfan colloidal gold ICA strip. (**A**) Pear: 0, 20, 40, 80 ng/g; (**B**) apple: 0, 20, 40, 80 ng/g.

**Table 1 foods-12-00736-t001:** Antisera detection results of endosulfan from the mice.

Mouse Number	Antisera Detection of the Third Immunization			Antisera Detection of the Fourth Immunization			Antisera Detection of the Fifth Immunization		
	Antisera Dilution Factor	Concentration of the Coating Antigen ^a^	IC_50_ ^b^	Antisera Dilution Factor	Concentration of the Coating Antigen ^a^	IC_50_ ^b^	Antisera Dilution Factor	Concentration of the Coating Antigen ^a^	IC_50_ ^b^
M-1	32 K	0.3	195.43	64 K	0.1	75.47	128 K	0.1	54.23
M-2	32 K	0.3	221.45	64 K	0.1	114.35	64 K	0.1	87.67
M-3	4 K	1.0	532.14	-	-	-	-	-	-
M-4	32 K	0.3	214.53	64 K	0.1	98.47	128 K	0.1	76.51
M-5	16 K	1.0	289.21	32 K	0.3	178.43	32 K	0.3	112.35

Note: ^a^ μg/mL; ^b^ ng/mL.

**Table 2 foods-12-00736-t002:** Optimization of the pH, methanol content and the NaCl content on endosulfan, through the ic-ELISA assay.

	pH				Methyl Alcohol Content					NaCl Content				
	6.0	7.2	8.5	9.6	0%	5%	10%	15%	20%	0%	0.4%	0.8%	1.6%	3.2%
A_max_	1.42	1.49	1.36	1.04	1.43	1.54	1.40	1.37	1.17	1.57	1.46	1.39	1.20	0.98
IC_50_	7.98	5.55	6.70	8.84	6.65	5.67	6.77	7.20	7.95	5.66	5.35	5.90	7.89	10.80
A_max_/IC_50_	0.18	0.27	0.20	0.12	0.22	0.27	0.21	0.19	0.15	0.28	0.27	0.24	0.15	0.09

**Table 3 foods-12-00736-t003:** The CR values of mAb using the ic-ELISA method.

Analytes	IC_50_ (ng/mL)	CR (%)
endosulfan	5.16	100.00
heptachlor	144.56	3.56
toxaphene	>1000	<0.1
chlordane	>1000	<0.1

**Table 4 foods-12-00736-t004:** Recovery rates of endosulfan in the pear and apple (*n* = 3).

Samples	Spiked Level (ng/g)	Recovery (%)	CV (%)
pear	5.0	113.45 ± 7.69	6.78
	10.0	91.48 ± 4.75	5.19
	15.0	105.19 ± 8.71	8.28
apple	5.0	92.39 ± 6.74	7.30
	10.0	98.54 ± 4.25	4.31
	15.0	106.12 ± 6.78	6.39

**Table 5 foods-12-00736-t005:** GC-MS analysis of the spiked samples (a: pear; b: apple).

Amount of Endosulfan Added (ng/g)	a GC-MS (ng/g)	Recovery ± SD (%)	b GC-MS (ng/g)	Recovery ± SD (%)
0	0	——	0	——
20	17.86 ± 1.12	89.3 ± 5.6	16.96 ± 1.06	84.8 ± 5.3
40	36.6 ± 1.36	91.5 ± 3.4	37.46 ± 1.84	93.65 ± 4.6
80	75.53 ± 2.56	94.41 ± 3.2	76.59 ± 2.8	95.74 ± 3.5

**Table 6 foods-12-00736-t006:** Comparison of this method with other measurement methods.

Reference	LOD (ng/mL)	Method	Analysis Time (h)	Detection Object
This work	40	ELISA and colloidal gold immunochromatographic strip	0.25	Apple and pear
6	8000	GC-MS/MS	1–2	Human biological samples
7	900	GC-MS	1–2	Natural soil and water
8	3	SPE LC-APCI/MS/MS	1–2	Sediments/soils, floating and submerged algal mats, and small fish
9	0.05	HPLC-UV	0.13	Environmental water
10	63	HPLC	1–2	Goat milk
15	0.8 (ng/g)	Enzyme-linked immunosorbent assays	5	Agricultural Products
16	1	Enzyme-linked immunosorbent assay	5	Fruits and vegetables

## Data Availability

All data and materials generated or analyzed during this study are included in this published article.

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
