# Peer review of "Development of Indirect Competitive ELISA and Colloidal Gold Immunochromatographic Strip for Endosulfan Detection Based on a Monoclonal Antibody"

_foods, 2023, doi:10.3390/foods12040736_

Round 1
Reviewer 1 Report
The authors have developed the first immunochromatographic assay for the insecticide endosulfan. They prepared mAb with high sensitivity and affinity against endosulfan, and then they developed ic-ELISA and colloidal gold ICA strip based on the prepared mAb, and they were applied for the rapid determination of endosulfan. The manuscript provides a quick visual tool for the detection of endosulfan in food samples; however, it should be revised prior.
① The authors should compare the performance of their method with previously reported methods for endosulfan (references 6 to 10 and 15 to 17) in tabular form, and the comparison parameters could include LOD, used technique, analysis time, etc.
② Why the authors used two protein carriers (OVA and BSA) for the hapten
③ How the authors prevented the non-selective absorption of Au-labeled antibodies on the NC membrane? Did they coat it with BSA or any reactive site blocker?
④ In section 2.9, what does the well contain, and how was the sample extraction solution prepared?
⑤ Please mention what the absorbent pad is made from.
⑥ The grammar in section 2.5 needs to be checked and corrected.
⑦ Please remove the abbreviation END throughout the manuscript, as it is confusing.
⑧ Line 209, correct “con-ditions”
⑨ Line 223, correct “figure figure5”
⑩ In lines 278-281, the format of the text is not according to the journal guidelines. Please correct.
Author Response
请参阅附件。

Reviewer 2 Report
This paper developed an indirect competitive enzyme-linked immunosorbent assay (ic-ELISA) and colloidal gold immuno- chromatographic (ICA) strip based on prepared monoclonal antibody (mAb) for quantitative and qualitative detection of endosulfan, with an mAb high sensitivity .this ic-ELISA showed a 50% inhibition concentration (IC50) value of 5.16 ng/mL for endosulfan.
I noted the in this paper is inserted the crossover rates of antibodies for few organochlorine insecticide but lack aldrin, dichloro-diphenyl-richloroethane, dieldrin, endrin or lindane, that are the most used in agricolture. it is possible inserted one of these.
The average recoveries of endosulfan in spiked samples (pear and apple), are good. it is possible insert another sample in which this pesticides is used as potatoes and tomatoes.
the paper lack of the comparison with analytical results reported in literature for other elisa method
Round 2
Reviewer 1 Report
The authors improved their manuscript adequately and it could be accepted for publication